# Geographical Origin Does Not Modulate Pathogenicity or Response to Climatic Variables of *Fusarium oxysporum* Associated with Vascular Wilt on Asparagus

**DOI:** 10.3390/jof7121056

**Published:** 2021-12-09

**Authors:** Alexandri María Brizuela, Justyna Lalak-Kańczugowska, Grzegorz Koczyk, Łukasz Stępień, Michał Kawaliło, Daniel Palmero

**Affiliations:** 1Department of Agricultural Production, Escuela Técnica Superior de Ingeniería Agronómica, Alimentaria y de Biosistemas, Universidad Politécnica de Madrid, 28040 Madrid, Spain; alexandri.brizuela@alumnos.upm.es; 2Plant-Pathogen Interaction Team, Institute of Plant Genetics of the Polish Academy of Sciences, Strzeszyńska 34, 60-479 Poznań, Poland; jlal@igr.poznan.pl (J.L.-K.); lste@igr.poznan.pl (Ł.S.); 3Biometry and Bioinformatics Team, Institute of Plant Genetics of the Polish Academy of Sciences, Strzeszyńska 34, 60-479 Poznań, Poland; gkoc@igr.poznan.pl (G.K.); mkaw@igr.poznan.pl (M.K.)

**Keywords:** asparagus officinalis, field decline syndrome, temperature, water activity, phylogenetics

## Abstract

Asparagus crop is distributed worldwide, covering very different climatic regions. Among the different diseases that affect asparagus, vascular *Fusarium* wilt, caused by *Fusarium oxysporum* f. sp. *aparagi* (*Foa*), stands out. It is not only the cause of large economic losses due to a decrease in yield and shortened longevity of the plantation, but also prevents replanting. This work aimed to determine if *F. oxysporum* isolates associated with vascular wilt on asparagus have adapted differentially to the different agro-environmental conditions. The potential correlation between origin and mycelial growth under different temperatures and humidity conditions was analysed for isolates from asparagus fields cultivated in northern and southern Europe. The genetic and pathogenic variability were also analysed. While a clear effect of water activity on mycelial growth was observed, all isolates responded in a similar way to changes in water activity in the medium, regardless of their geographical origin. The results revealed a low genetic variability of *F. oxysporum* isolates associated with vascular wilt on asparagus without signs of differentiation correlated to geographical origin. The southernmost isolates of the two cultivated varieties inoculated did not express more pathogenicity than those isolated from the colder region.

## 1. Introduction

Asparagus is the fourteenth largest open-field horticultural crop in the world, with more than 1.6 million hectares dedicated to it. In Europe, where more than 320,000 tons are produced annually across over 62,191 hectares, its geographical distribution covers the entire span of the continental territory [1]. Spain and Poland are the second and sixth largest asparagus producers in Europe, with 58,610 and 12,700 tons, respectively [1]. In the case of Spain, exports are around 24,701 t with a total value of 70.4 million euros [2].

However, the crop has experienced a substantial decrease in yield in recent years due to fungal diseases [3]. Among the key diseases affecting this crop, vascular wilting of asparagus stands out [4,5,6] worldwide. This disease, together with the appearance of root rot, causes what has been defined as Asparagus Decline Syndrome (ADS) [7], which is characterised by the significantly shortened longevity of the crop and a limiting factor in economic production.

The disease was first described by Cohen and Heald [8] who identified *Fusarium oxysporum* f. sp. *asparagi* as the pathogen that caused it. It is a complex syndrome in which biotic and abiotic aspects are intertwined [9,10]. The autotoxicity of asparagus predisposes the roots to soil-borne fungi. *F. oxysporum* f. sp. *asparagi* is the species that seems to prevail as the most pathogenic within the specific complex associated with the disease. Several studies have extended its aetiology to a complex of *Fusarium* species, which is highly variable depending on the geographical regions that have been associated with the syndrome [3,4,11,12,13,14]. During the productive period, no symptom other than a decrease in production is observed. It is only at the end of the growing cycle when the plants are allowed to vegetate to accumulate reserves that premature yellowing of plants is observed, and finally the whole plant wilts.

It is known that stressors increase the incidence and severity of the disease [9], and different studies have delved into the identification of the source of the inoculum [15]. Regarding disease control, planting methods, irrigation and fertilisation regimes, and pest management practices have been previously evaluated to determine their role in the predisposition of asparagus to Fusarium disease [15]. There have also been studies on the use of salt [16] to increase the resistance of asparagus plants, as well as the use of non-virulent isolates of *F. oxysporum* [8,17] or arbuscular-mycorrhiza to induce systemic resistance [18,19]. Although different authors have approached the search for resistant plant material [20,21], the available asparagus cultivars have low resistance, and most authors agree that cultural practices that reduce stress are recognised as the best practical method to prevent the decline caused by *Fusarium* spp.

It seems that the pathogen enters the plant through the secondary roots and passes to the storage roots [22]. Moreover, the perennial nature of asparagus cultivation together with the harvesting method forces the emerging shoots to be cut every few days at ground level. This causes wounds that directly expose the vascular tissues of the plant to soil microorganisms, facilitating the colonisation of the crown of the asparagus plant. These are factors that have not yet been solved for effective control of the disease. Although there are products that have some effects under controlled conditions [23], continuous harvesting and the absence of aerial parts during the entire productive period of the plant makes the application of phytosanitary products very difficult.

On the other hand, in recent decades there has been a notable increase in global temperature, significantly influencing the soil pathosystems [24], especially in the southernmost areas but also in colder areas. Although it is known that the different pathogenic species vary with the climatic conditions, there have been no studies on whether the same species has been able to adapt to different agro-environmental conditions.

Over the last decade, a variety of approaches have been taken to examine *F. oxysporum* using various physiological and molecular methods, such as vegetative compatibility groups (VCGs) [25], random amplified polymorphic DNA markers (RAPDs; [26]), restriction fragment length polymorphisms (RFLPs) [27,28], amplified fragment length polymorphisms (AFLP) [29], and DNA sequence analyses [30]. Moreover, a multilocus sequence-typing (MLST) database, *Fusarium* MLST (http://www.cbs.knaw.nl/fusarium/ [31]), with partial DNA sequences from *TEF1*, *RPB1*, and/or *RPB2*, was developed to compare genes from individual phylogenetic *Fusarium* species. Furthermore, assessments of intraspecies diversity and phylogenetic relationships within *Fusarium* species from various regions have provided information regarding the pathogen’s spread across different geographical areas and may clarify the appearance as well as evolution of this plant pathogen. Such a deep understanding of the population’s biogeographical structure as well as the relationships within the varied representatives of *F. oxysporum* plant pathogens is essential for the successful development of new disease management strategies and for effective resistance breeding [32].

The present study aimed to describe the possible adaptation of *F. oxysporum* isolates associated with vascular wilt on asparagus from two producing regions of Europe. Assessing the level of genetic variability within the pathogen populations is significant as a high genetic variation indicates a fast evolution in response to changing ecosystems [33]. Therefore, the genetic variability of *F. oxysporum* isolates was assessed to evaluate whether the genetic differentiation relates to geographic distribution. The effects of temperature and water activity on the mycelial growth of isolates from the northern and southern European production regions were also assessed in order to better understand the response to changing environmental conditions. The virulence of selected isolates was assessed by inoculation into two asparagus cultivars.

## 2. Materials and Methods

### 2.1. Isolates

A total of 46 *Fusarium oxysporum* strains (Table 1) isolated from vascular tissues of decayed asparagus plants in different fields in three EU countries (20 from Spain, 21 from Poland, and 5 from The Netherlands) were used in this study. The analysis of the plant samples consisted of the superficial disinfection of secondary and storing roots with 1.5% sodium hypochlorite solution for 1 min, followed by two successive washings with sterile distilled water. After drying, 1 cm pieces were sown in plates with a potato dextrose agar (PDA) culture medium supplemented with 0.5 g/L of streptomycin sulphate (Sigma-Aldrich, St. Louis, MO, USA) (PDAS) and incubated for 5–7 days at laboratory temperature (25 °C) under continuous fluorescent light. Fungal single-spore cultures were obtained from the different *Fusarium* colonies recovered. Isolations were carried out in the country of origin where the strains were properly stored in fungal collections on potato dextrose agar (PDA; Merck, Darmstadt, Germany) medium at 4 °C in the dark. Some of the Spanish strains had already been investigated in a previous study [3] and were included for further characterisation in a wider geographical context.

### 2.2. DNA Extraction, Primers, PCR Assays, and DNA Sequencing

The genomic DNA of all of the isolates were extracted following the hexadecyltrimethylammonium bromide (CTAB) method previously described by Stępień et al. [34] and were used as a template for PCR amplification at the following conditions: initial denaturation at 95 °C for 3 min, 35 cycles of denaturation at 95 °C for 30 s, primer annealing at 59 (for *EF-1α*), 57 (for *RPB1*) or 55 °C (for *RPB2*) for 20 s, and final extension at 72 °C for 60 s. Amplification reactions were performed in volumes of 25 μL containing 0.1 μL of each primer (100 μM), 0.5 μL of DreamTaq Green DNA polymerase (Thermo Scientific, Espoo, Finland) (5 U/μL), 2.5 μL of 10× PCR buffer, 1.5 μL of dNTPs (10 μM), and 10 ng of genomic DNA as template.

Specific primers for the amplification of partial sequences of the translation elongation factor-1α (*EF-1α*), and the DNA-directed RNA polymerase II largest (*RPB1*) and second largest subunit (*RPB2*) genes of *Fusarium oxysporum* were used as follows: ef1—forward primer 5′-ATGGGTAAGGARGACAAGAC-3′; ef2—reverse primer 5′-GGARGTACCAGTSATCATGTT-3′ [30]; Fa—forward primer 5′-CAYAARGARTCYATGATGGGWC-3′ [35], R8—reverse primer 5′-CAATGAGACCTTCTCGACCAGC-3′ [31]; —forward primer5 f2 5′-GGGGWGAYCAGAAGAAGGC-3′ [36], and 7cR—reverse primer 5′-CCCATRGCTTGYTTRCCCAT-3′ [37].

Amplified DNA fragments were electrophoresed in 1.5% agarose gels (Prona Agarose; ABO, Gdańsk, Poland) in a 1X concentrated TAE (Tris-acetate-EDTA) buffer (Fermentas, Vilnius, Lithuania) with GelGreen Nucleic Acid Stain (Biotium, Inc., Vladimir, Russia) and were visualized over an ultraviolet transilluminator.

The generated PCR amplicons were purified using exonuclease I (Thermo Scientific) and shrimp alkaline phosphatase (FastAP, Thermo Scientific) according to [34]. Sequencing reactions were carried out using the BigDye Terminator v3.1 kit (Applied Biosystems, Foster City, CA, USA) and sequence reading was performed using ABI 3730 DNA Analyzer (Applied Biosystems, Waltham, MA, USA) equipment.

### 2.3. Sequence Alignment and Phylogenetic Analysis

The obtained amplicon sequences were compared against reference genes and coding sequences (*TEF1A*, *RPB1*, and *RPB2*) for model fusaria (*F. oxysporum* Fo47 [38] and *F. proliferatum* ET1 [39]), as identified in Ensembl/Fungi [40] based on a BLASTP search of translated sequences with *Fusarium proliferatum* exemplars annotated in the UniProt database.

In order to preserve intron–exon boundaries for partitioned phylogenetic analysis, firstly, the alignments of the reference coding sequences were obtained on the basis of backtranslated protein alignments (MAFFT v 7.407 [41]; EINSI algorithm; backtranslation with trimal [42]). Afterwards, gene and amplicon sequences were added to this backbone using MAFFT’s -addlong and -addfragment options. Exon–intron boundaries were inspected and corrected as needed (based on known splice site sequences). Finally, the cDNA references were removed from the final alignments, and the alignments were trimmed to cover only the sequenced amplicon regions.

Partitioned, maximum likelihood, and phylogenetic analyses were carried out in IQTREE v2.1.4-beta [43] based on automated model detection for nucleotide substitutions. As no parsimony informative sites were observed in the coding parts of *TEF* amplicons, the *TEF* exonic sequence was not included among the analysed partitions. The remaining *RPB1*, *RPB2*, and *TEF* intron sequences were ascribed separate partitions (see Appendix A for individual alignments, as well as the IQTREE partition file in NEXUS format). The resulting consensus phylogenetic tree (233 ultrafast bootstrap iterations) was visualised via a custom ETE2 [44] Python script (*F. proliferatum* ET1 outgroup; collapsed branches with less than 50% support).

### 2.4. Analysis of the Effect of Climate Variables on the Mycelial Growth of Isolates

The response of the different isolates to environmental variables such as temperature and water activity was evaluated on six selected isolates of *F. oxysporum*, three of them from Spain (coded as FOA02, FOA06, and FOA09) and three from Poland (coded as GA1.4, GASP99.2, and GA1.2). To determine the interactive effects of temperature and water activity on the growth of isolates, we used a 6 × 4 × 4 × 5 completely randomised factorial experimental design, wherein strains 1 to 6 were the first factor, temperature was the second factor (15–20–25–30 °C), and water activity (a_w_) was the third factor. Potato dextrose agar (PDA; Merck, Darmstadt, Germany) was used in this study. The value of a_w_ was modified with the non-ionic solute glycerol to yield final a_w_ values of 0.99, 0.97, 0.95, and 0.93. The solutes were not added to the control medium (a_w_ = 0.996). The a_w_ of the media was checked with a hygrometer (AquaLab 3TE; Decagon Devices Inc., Pullman, WA, USA). Each combination of isolate, temperature, and water activity was replicated five times. Mycelial discs 5 mm in diameter were excised from the edge of 8-day-old fungal colonies. Inoculated plates were incubated for 5 days. Two perpendicular straight lines were drawn on the bottom of each Petri dish. The crossing point coincided with the centre of the 5 mm initial fungi disc. The radial growth of the pathogens was measured daily with a digital Vernier following the four segments formed by the two perpendicular lines. Data for each day corresponded to the means of four measurements. Bioassays were ended after two weeks when the fungi mycelia reached the Petri dish wall. Daily fungal growth rate was calculated for each fungi matrix and expressed as mm·d^−1^.

### 2.5. Pathogenicity Test on Plants

Conidial suspensions were adjusted to approximately 10^6^ conidia mL^−1^ and used to inoculate healthy asparagus seedlings after cultivation in sterile (autoclaved twice at 105 kPa for 30 min) soil for five weeks. Two different commercial varieties selected with different degrees of tolerance to disease [X] (cv. Vegalin and cv. Grande, susceptible, and moderately tolerant to *Fusarium* spp., respectively) were used for the tests. Seedling roots were soaked in the conidial suspensions of each *Fusarium* isolate for 24 h before planting in flats containing sterile soil. Three replicate flats were prepared for each isolate and each asparagus cultivar. The plants were maintained in a temperature and light-controlled greenhouse (12 h/12 h light/dark; 25/21 °C). All of the treatments were replicated four times. Disease symptoms were graded into five classes following the method of [45], as follows: 1 = no symptoms; 2 ≤ 10% rotted roots; 3 = 10–50% rotted roots; 4 > 50% rotted roots; 5 = completely rotted roots. A Disease Severity Index (DSI) for each *Fusarium* isolate was calculated as the mean of severity estimated for three plants of each variety based on four technical replicates. Symptoms on asparagus plants were recorded three weeks after inoculation. At the end of the experiment, the dry weights of the plants were recorded following oven-drying at 80 °C for 48 h.

### 2.6. Statistical Analysis of Data

An analysis of variance on the Disease Severity Index (DSI) and dry weight from the pathogenicity test with *Fusarium oxysporum* isolates as a random factor depending on the origin (Spain and Poland) and inoculated cultivar, as well as on the effects of temperature and water activity as fixed predictors on the growth rate (response) of *Fusarium* isolates were performed using Fisher’s least significant difference (LSD) tests at 99.9% confidence. The tests were carried out using STATGRAPHICS Centurion XVIII statistical package software (StatPoint, Inc., Herndon, VA, USA). In order to estimate the correlation between the growth rate at any of the studied temperatures and the water activity, the simple regression analysis was adjusted to the non-linear inverse-X model as it showed the highest R2 value.

The same statistical package software was used to construct a generalised linear model of the growth rate (Yi = β0 + β1X1,i + β2X2,i + β3X3,i + … + βkXk,i + εi, where “Y” is the response variable (growth rate), “β1Xk,i” the predictor variables (temperature and water availability), and “i” the error). The linear regression of increase in radius against time (d) was used to obtain growth rates (mm·d^−1^) as indicated in 2.5 for each set of treatments. Pearson correlation coefficient at a 5% confidence level was used as significance criterion.

## 3. Results

### 3.1. Phylogenetic Analysis

The maximum-likelihood (ML) phylogeny derived from the concatenated loci is shown in Figure 1. The phylogenetic analysis resulted in two well-supported clades, delineated based on FOA09 (isolate from Spain) and GASP4 (isolate from Poland), respectively, which have highly supported sub-clades, although their genetic divergence is relatively low. Clade I was the most frequent group and included isolates from three EU countries with a strong bootstrap support of 86%. Clade II consisted of five isolates of *Fusarium oxysporum* from Spain (south subpopulation) and three isolates from the Netherlands (north subpopulation) with strong bootstrap support of 84%. The combined gene tree reveals that *F. oxysporum* isolates from Spain are identical to the isolates from the Netherlands and Poland. Numerous isolates from the same geographic population are grouped together. Indeed, *F. oxysporum* isolates were distributed across the dendrogram, and their clusters did not show a correlation with the geographical asparagus production regions in which the isolates were obtained. However, there were Spanish isolates that grouped with the isolates from the Netherlands or Poland.

### 3.2. Response to Temperature and Water Activity

The possible differential response of the isolates’ mycelial growth against temperature was studied at four different temperatures. The effect of temperature on the mycelial growth of *Fusarium oxysporum* associated with vascular wilt on asparagus was evaluated on Potato Dextrose Agar (PDA) media. Our experimental results revealed that this pathogen grows well at temperatures in the range of 25 to 30 °C, with preference towards the lower temperature (25 °C), and also demonstrated a clear increase in growth as the water activity of the medium increased (Figure 2).

The dependence of the mycelial growth rate on geographic origins was also analysed (Figure 3). Isolates from a different origin do not react differently to water under certain temperatures, the ANOVA three-way interaction between the origin, water activity, and temperature did not show significant differences (*p* = 0.1544), neither did second order with aw and temperature (*p* = 0.1192 and *p* = 0.4134, respectively) nor third order interactions (*p* = 0.8559). Indeed, no direct correlation was detected except for the experiment part carried out for a combination of both higher temperature and higher water activity, where the Spanish isolates showed slightly higher growth rates than the Polish ones (*p* = 0.0154).

Taken together, the above results show that although a clear effect of water activity on mycelial growth was observed, in general, the response to changes in water activity did not vary significantly between the analysed isolates (Figure 4).

With regard to the temperature factor, the regression analysis showed statistically significant *p*-values (*p* < 0.001) for all of the temperatures tested. In this context, high R^2^ coefficients (ranging from 82.22% to 93.65%) expressed the proportion of the variation of the radial growth rate that is explainable by a_w_ and T (Figure 4 and Table 2). Within the a_w_ and temperature ranges specified, the selected models could accurately predict the fungal growth rates (mm·d^−1^), in agreement with previous work on other special forms [46].

The GLM method allowed us to estimate the repeatability and reproducibility of the fungal growth measurement, i.e., R-squared = 84.75% (adjusted R-squared = 84.26%). After the stepwise variable selection, three effects were selected in the model including temperature cultivation, water activity, and second order interaction (growth = −28.794 − 1.431 × Tre + 31.5442 × a_w_ + 1.55464 × Tre × a_w_). The joint effects of water activity and temperature on growth coefficients are visualised as a surface plot in Figure 5. The highest growth rates were observed when high temperatures were combined with high water activities. The contour levels revealed a peak of the growth rate (more than 5 mm/day) experiment when the temperature was more than 25 °C and water activity was over 0.997.

### 3.3. Pathogenicity Study

Tests conducted on asparagus showed the pathogenic capacity of all *Fusarium oxysporum* strains inoculated (Table 3). All of the isolates produced symptoms with disease scores significantly different from those of the controls (*p* < 0.001) on the two inoculated varieties. Three weeks after the inoculation of asparagus, symptoms included necrosis upon the insertion of feeding rootlets with storage roots and water-soaked rotten roots, which eventually disintegrated and progressed into the vascular tissues in accordance with that described in [3,47,48]. Disease Severity Index (DSI) values from the inoculated asparagus varieties were significantly greater than those in the controls (Table 3). The highest DSIs were recorded after inoculations with isolates FOA02 and GA1.4, with DSIs over three points higher than those of the un-inoculated control seedlings. No differences were observed for the dry weight of the aerial tissues and root.

Overall, there was no significant difference in susceptibility between the two inoculated cultivars (*Vegalin* and Grande) or in response to the origin of the isolates (North and South EU) nor for the second order interactions (*p* > 0.05). There was no differential varietal response (*p* > 0.05) that corresponded to previous studies [12,49], which, at the moment, did not allow us to indicate the use of resistant varieties as an effective method to control the disease in the field. 

## 4. Discussion

*Fusarium oxysporum* isolated from asparagus has been studied for many years [15,16]; previous results revealed epidemiological information about *F. oxysporum* in asparagus fields, describing the effects of ecological factors such as the temperature on the density of *F. oxysporum* inoculum in the soil [3]. However, the effects of some other factors on fungal growth, such as water activity or temperature, remain to be elucidated, and little is known about the pathogen’s genetic variability. Moreover, no current studies have been conducted to examine the potential relationship between genetic variability and geographical location of *F. oxysporum* isolates from different European regions. The above experimental results demonstrate that the *F. oxysporum* population associated with vascular wilt on asparagus likely does not show variability according to geographical regions separated by more than 2000 km distance (Spain, The Netherlands, and Poland). This could be explained by the considerable freight traffic resulting in infected asparagus runner plants transplanted in fields that could come, for instance, from Spanish nurseries, but also from other production regions such as the Netherlands and Poland. An additional factor that could explain the lack of differences in variability between isolates from the three plant-growing countries is the tendency towards asparagus cultivation as a monoculture. Nevertheless, in spite of low variability, isolates from different production regions formed groups in phylogeny reconstruction. This points to the geographical origin not being a strong predictor of common descent, despite the different environmental conditions, different disease management practices being followed, the use of different asparagus varieties, or even the different history of cultivation [3]. In contrast with our study, Wong and Jeffries [50] stated a spread across a number of clades of *F. oxysporum* isolates that are pathogenic to asparagus roots from two different production regions—Spain and UK—within the species complex. They concluded that pathogenicity to asparagus roots in this species is a quite unspecialized trait.

As a sidenote, while our investigation revealed phylogenetic relationships between the investigated isolates, it was based on three reliable taxonomic markers rather than entire genomes. The employed housekeeping genes thus provide only a lower bound, conserved assessment of intraspecies variability. The individual strains likely harbour more significant diversity within genome components under weaker selective pressure (in particular supernumerary chromosomes).

The experimental results obtained highlight a significant effect of a_w_, temperature, and the interactions of these factors on the radial growth of *F. oxysporum* isolates. Notably, the mycelial growth of isolates of different special forms of *Fusarium oxysporum* has been investigated in previous works [51,52,53], and the optimal temperature ranges are very similar for the different special forms studied, where the mycelial growth rate was greater at 23/26 °C. In some cases, the mycelial growth rate was not directly studied and only the manifestation of symptoms was monitored following inoculation (in different temperature ranges). There, cotton seedlings manifested more severe symptoms and higher mortality when maintained at 23 °C, followed by 20/26 °C and 29 °C in descending order [51]. Furthermore, Scott et al. [54] in their study on the *Fusarium* vascular wilt of lettuce caused by *F. oxysporum* f. sp. *lactucae* indicated that the tendency for more severe disease under warmer conditions may be partly due to an effect of temperature on the growth of the pathogen up to an apparent maximum near 25 °C, which is coincident with our results on mycelial growth on *F. oxysporum* isolated from diseased asparagus plants.

Therefore, it does not seem that the origin of the isolates allowed for an adaptation to the climate that corresponds to a better response to high or low temperatures. Rather, the results, together with those of the phylogenetic analysis, make it possible to advocate a greater uniformity of the *F. oxysporum* isolates associated with the vascular wilt of asparagus without major differences due to their origin. However, it is important to note that the altitude of the areas of origin of the Spanish isolates (between 600 and 900 m above sea level) compared with the low altitude of the Polish isolates (less than 200 m above sea level) could have partly compensated for the different latitudes.

Several studies have highlighted the importance of predictive microbiology as a tool for developing mathematical fungal growth models. Pertinently, such models facilitate the prediction of microbial outbreaks in relation to abiotic factors [55,56]. However, it should be pointed out that the model outlined above is based on data obtained under in vitro conditions. Thus, the predicted growth could be faster than growth under natural conditions due to the nutrient richness in the artificial medium [57].

The exchange of plant material has been indicated as one of the main routes of entering of the pathogen into new areas [3]. The high pathogenic capacity observed in the isolates from such geographically remote areas leads us to emphasise the importance, highlighted in previous works, of avoiding plant material to introduce an inoculum together with planting material from areas with asymptomatic plants [15,48]. In a previous work [3], we revealed epidemiological information about *F. oxysporum* in asparagus fields, describing the effects of ecological factors such as the temperature on the density of *F. oxysporum* inoculum. The effects of some other factors, such as water activity, remained to be elucidated; thus, we attempted to do so in this study. Although the difficulty of translating these in vitro growth results to the field is obvious, the experimental results allowed for understanding the relationship between the two main factors studied and the mycelial growth response values, setting up the stage for the future investigation of regulatory and epigenetic determinants.

## 5. Conclusions

Taken together, our results demonstrate a low genetic variability of *Fusarium oxysporum* species associated with asparagus. Further genetic studies of this or more extensive populations using deeper genomic coverage can be conducted to better understand *F. oxysporum* populations associated with asparagus and their capacity to overcome deployed disease management strategies and to support international asparagus producers.

If we take into account that the water activity is directly related to available water, on the basis of our results regarding the influence of water activity on the growth capacity of these pathogens, it is possible to explain the entry of pathogens in the first years of cultivation. The colonisation process is likely favoured when the crop is irrigated excessively, it can also be associated with heavily or easily flooded soils where the water activity is naturally high for long periods of time. These results should alert growers of the influence of the climate change on the sanitary status of the asparagus cultures. The practical implementation of these results can be used to facilitate field monitoring and will contribute to farmers adopting more rational control strategies for new asparagus plantations.

## Figures and Tables

**Figure 1 jof-07-01056-f001:**
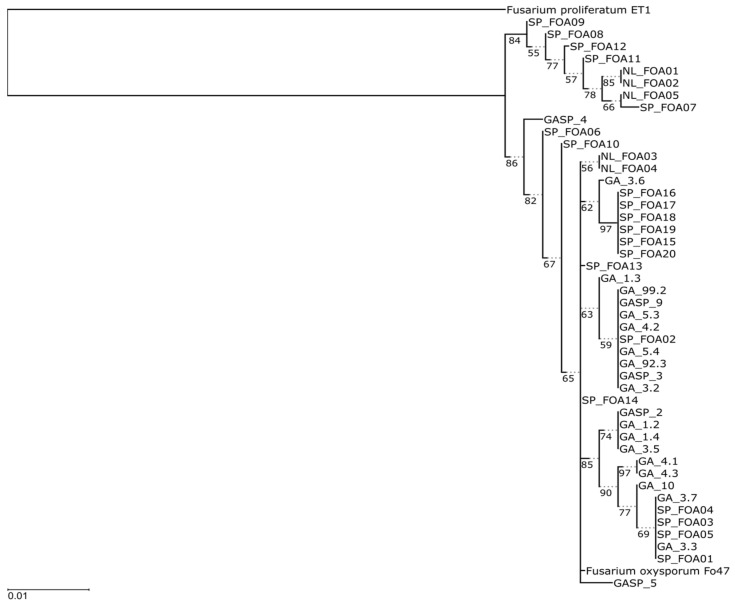
Consensus tree based on the maximum-likelihood analysis of the partitioned alignment of amplicons (partitions: RPB1/TNe+I/915 bp, RPB2/TNe+I/853 bp, TEF1A intron #1/K2P+R2/184 bp, TEF1A intron #2/K2P/55 bp; 22 parsimony informative sites across all partitions). Branches with less than 50% support were collapsed. Solid lines drawn to scale (units of change), dotted lines represent padding.

**Figure 2 jof-07-01056-f002:**
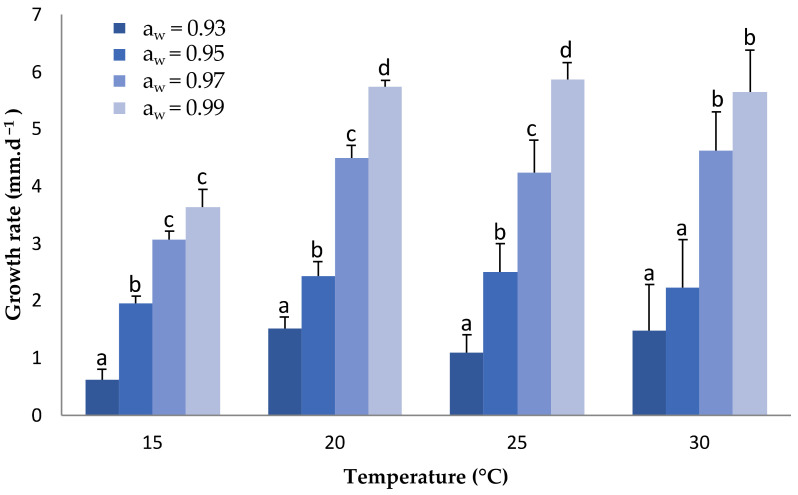
Average mycelial growth of isolates of *F. oxysporum* associated with vascular wilt on asparagus at different regimes of temperature and water activity (a_w_). Bars with the same lower case letter did not differ significantly *p* ≤ 0.05.

**Figure 3 jof-07-01056-f003:**
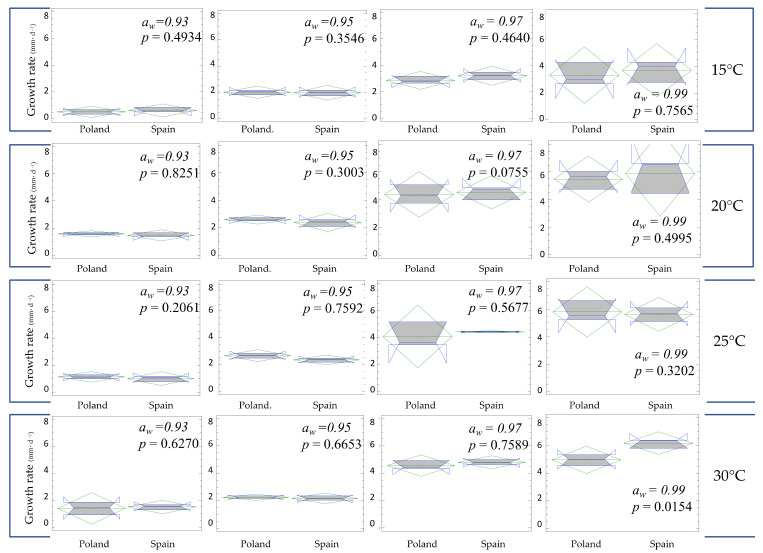
Box plots showing mycelial growth rate of *Fusarium oxysporum* associated with vascular wilt on asparagus at different regimes of temperature and water activity (a_w_) depending on the cultivation region. The right of the box is the 75th percentile; the left is the 25th percentile; and the whiskers represent the maximum and minimum values.

**Figure 4 jof-07-01056-f004:**
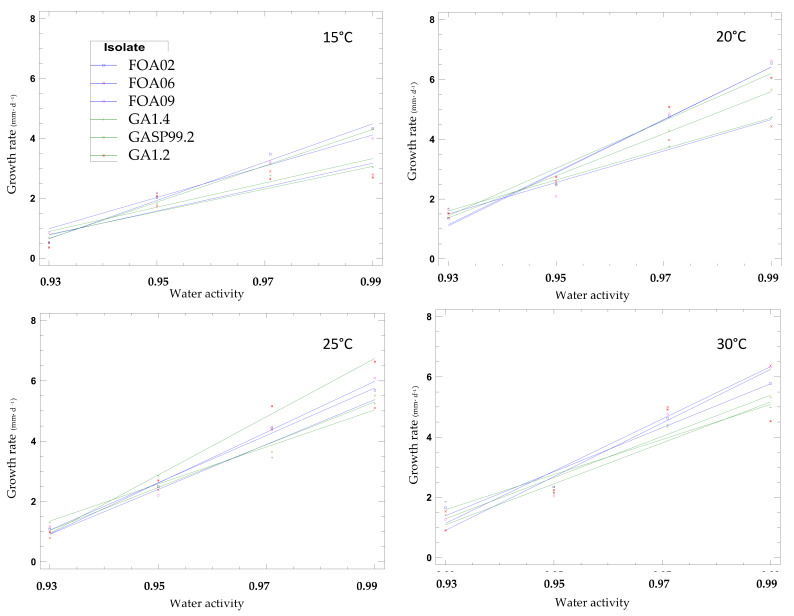
Growth rate of the different isolates of *F. oxysporum* from Poland (**green line**) and Spain (**blue line**) analysed at the different temperatures and water activities tested.

**Figure 5 jof-07-01056-f005:**
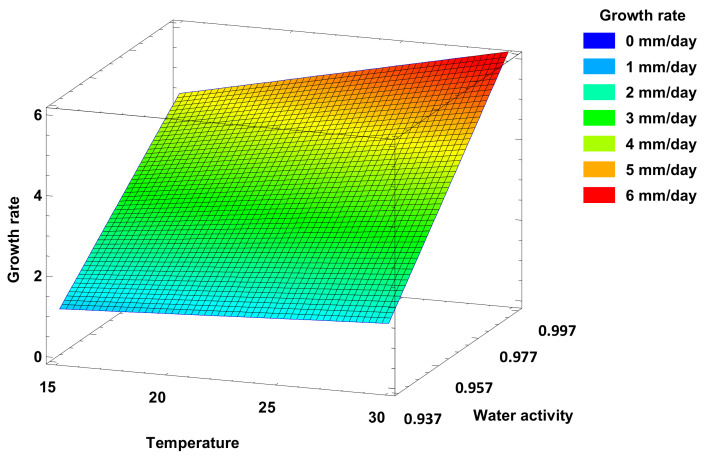
Surface contour plot of growth rate responses at varying temperatures and water activities.

**Table 1 jof-07-01056-t001:** Origin of *Fusarium oxysporum* isolates used in this study.

Origin	N	Country Code	Isolates Codes	Isolate Source
Spain	20	SP	01 to 20	Asparagus plant
Poland	21	GA	1.2, 1.3, 1.4, 3.2, 3.3, 3.5, 3.6, 3.7, 4.1, 4.2, 4.3, 5.3, 5.4, 99.2, 92.3	Asparagus plant
		GASP	2, 3, 4, 5, 9	
Netherlands	5	NL	01 to 05	Asparagus plant

**Table 2 jof-07-01056-t002:** Equations and correlation coefficients between growth rate (mm·d^−1^) and water activity (a_w_), evaluated for every tested temperature (T). Adjustment: inverse X function: Y = a + b/X.

T	N	Adjusted Equation	Correlation Coefficient	*p*-Value	R^2^
15 °C	120	Growth rate = 48.733–44.7756/a_w_	−0.910998	0.0000	82.22%
20 °C	120	Growth rate = 71.6285–65.6407/a_w_	−0.939446	0.0000	88.26%
25 °C	120	Growth rate = 77.4758–71.416/a_w_	−0.967739	0.0000	93.65%
30 °C	120	Growth rate = 72.8447–66.8726/a_w_	−0.934273	0.0000	86.71%

**Table 3 jof-07-01056-t003:** Disease Severity Index and dry weight of asparagus seedlings following artificial inoculation with six isolates of *F. oxysporum* associated with vascular wilt on asparagus originating from Spain and Poland.

**Variety**	**Grande**				**Grande**				**Variety**	**Grande**				**Grande**			
**Isolate**	**DSI root**				**DSI shoot**				**Isolate**	**Dry weight root**				**Dry weight shoot**			
	**Mean**	**±**	**SD**	**LSD**	**Mean**	**±**	**SD**	**LSD**		**Mean**	**±**	**SD**	**LSD**	**Mean**	**±**	**SD**	**LSD**
**Control**	0.2	±	0.28	a	0.1	±	0.14	a	**Control**	0.2	±	0.11	NS	0.3	±	0.01	e
**FOA02**	1.1	±	1.56	ab	0.7	±	0.42	ab	**FOA02**	0.17	±	0.12	NS	0.16	±	0.02	abc
**FOA06**	2.9	±	0.42	c	2.1	±	0.71	cd	**FOA06**	0.11	±	0.05	NS	0.2	±	0.03	cd
**FOA09**	3.4	±	0.85	c	2.4	±	0.28	cd	**FOA09**	0.02	±	0.01	NS	0.11	±	0	a
**GA1.4**	3.6	±	0	c	2.8	±	0.57	d	**GA1.4**	0.05	±	0.01	NS	0.18	±	0.03	bcd
**GASP99.2**	2.4	±	0.28	bc	1.5	±	0.14	bc	**GASP99.2**	0.1	±	0.01	NS	0.24	±	0.01	de
**GA1.2**	3.4	±	0.57	c	2.2	±	0.28	cd	**GA1.2**	0.05	±	0.05	NS	0.12	±	0.05	ab
**Significance**	***p* = 0.0146**				***p* = 0.0028**				**Significance**	***p* = 0.1945**				***p* = 0.0020**			
**Variety**	**Vegalin**				**Vegalin**				**Variety**	**Vegalin**				**Vegalin**			
**Isolate**	**DSI root**				**DSI shoot**				**Isolate**	**Dry weight root**				**Dry weight shoot**			
	**Mean**	**±**	**SD**	**LSD**	**Mean**	**±**	**SD**	**LSD**		**Mean**	**±**	**SD**	**LSD**	**Mean**	**±**	**SD**	**LSD**
**Control**	0.1	±	0.14	a	0.1	±	0.14	a	**Control**	0.15	±	0.05	b	0.26	±	0.01	d
**FOA02**	1.3	±	0.42	b	0.4	±	0	a	**FOA02**	0.12	±	0.02	b	0.16	±	0.02	c
**FOA06**	3.5	±	0.42	c	2.4	±	0.28	c	**FOA06**	0.02	±	0.02	a	0.13	±	0.03	bc
**FOA09**	3.8	±	0.28	c	2.8	±	0.28	c	**FOA09**	0.01	±	0.01	a	0.07	±	0.02	a
**GA1.4**	3.8	±	0.28	c	2.6	±	0.28	c	**GA1.4**	0.01	±	0	a	0.09	±	0.02	ab
**GASP99.2**	3.3	±	0.14	c	1.8	±	0.28	b	**GASP99.2**	0.05	±	0.03	a	0.13	±	0.02	bc
**GA1.2**	3.3	±	0.42	c	2.4	±	0	c	**GA1.2**	0.04	±	0.02	a	0.14	±	0.04	bc
**Significance**	***p* = 0.0000**				***p* = 0.0000**				**Significance**	***p* = 0.0053**				***p* = 0.0027**			
**Variety**	**DSI root**				**DSI shoot**				**Variety**	**Dry weight root**				**Dry weight shoot**			
	**Mean**	**±**	**SD**	**LSD**	**Mean**	**±**	**SD**	**LSD**		**Mean**	**±**	**SD**	**LSD**	**Mean**	**±**	**SD**	**LSD**
**Grande**	0.2	±	0.28	NS	0.1	±	0.14	NS	**Grande**	0.2	±	0.11	NS	0.3	±	0.01	NS
**Vegalin**	0.1	±	0.14	NS	0.1	±	0.14	NS	**Vegalin**	0.15	±	0.05	NS	0.26	±	0.01	NS
**Significance**	***p* = 0.6985**				***p* = 1.0000**				**Significance**	***p* = 0.6238**				***p* = 0.0880**			
**Origin**	**DSI root**				**DSI shoot**				**Origin**	**Dry weight root**				**Dry weight shoot**			
	**Mean**	**±**	**SD**	**LSD**	**Mean**	**±**	**SD**	**LSD**		**Mean**	**±**	**SD**	**LSD**	**Mean**	**±**	**SD**	**LSD**
**Spain**	2.67	±	1.26	NS	1.87	±	1.05	NS	**Spain**	0.07	±	0.08	NS	0.14	±	0.05	NS
**Poland**	3.3	±	0.52	NS	2.15	±	0.46	NS	**Polony**	0.05	±	0.03	NS	0.15	±	0.05	NS
**Significance**	***p* = 0.1227**				***p* = 0.4008**				**Significance**	***p* = 0.3381**				***p* = 0.6343**			

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
