# Peer review of "Geographical Origin Does Not Modulate Pathogenicity or Response to Climatic Variables of Fusarium oxysporum Associated with Vascular Wilt on Asparagus"

_jof, 2021, doi:10.3390/jof7121056_

Round 1

Reviewer 1 Report

The authors describe the phylogenetic relationships between Foa isolates collected in northern and southern Europe, they test if (selected) isolates respond differently to experimental water and temperature manipulations in bioassays and experimentally show the effects of the on two variaties of asparagus plants . As an ecologist I especially found the combinations of field observations and manipulations in the lab interesting and meaningful, while I expected other methodologies based upon the title and I have some open questions concerning the analysis of the data. Please find some comments to further improve the manuscript below.

It would be very helpful to the reader to formulate your three research questions at the end of the introduction. What are your expectations and do the results answer these questions? For example, is the genetic diversity of the fields assessed, or do you mean phylogenetic relatedness? What is pathogenetic diversity? shortly describe the experimental approach you use in the introduction and explain why you choose to test the pathogenicity of each of the isolates on 2 cultivars (explain the difference between the cultivars)

I also struggle a bit with the description of the methods. There seems to be no table 1 and table 2 comes after table 3 which is quite confusing for the reader. Please justify why you used partial sequences to construct the phylogenetic tree, are these primers not chosen for their conservativeness? Is it possible that the isolates differ more strongly in more functional parts of the genomes? How do the six selected isolates for the climate responses relate to the two clusters found in the phylogenetic tree?

I’m not familiar with the software used in this manuscript but from the description it remains unclear which predictors were included in the models and how the model was structured. It would be good to have a table with a statistical summary of the model -including origin (north south), temperature and water availability as fixed predictors and isolate as random factor on mycelium growth (response). I cannot understand what Figure 2 is showing and where these extremely high R2 values come from. Fig 4 is already clearer, showing increased growth rates with increasing water activity. To show interactions between the isolate origin and temperatures one could colour the isolates from the north blue and from the south orange. It would be good to also have a summary table for the tests of pathogenicity on asparagus plants. These probably should include origin (north south) and commercial variety (Grande Vegalin) fixed effects and isolate random and the responses could be DSI and dry weight.

In the discussion you mention that the genetic diversity of the tested pathogen is low resulting in similar responses along climatic gradients and similar disease effects of the isolates on asparagus plants. I tend to disagree with this, as you show the two clusters in the analysis as well as different responses for the isolates. I would strongly encourage the authors to redo the analysis and include origin into the models and resulting graphs. All in all I believe the study is topical and interesting but the manuscript needs some improvements before it can be plublished.    

Author Response

Rewiewer 1

Article ID jof-1443396 entitled "Geographical origin does not modulate Fusarium oxysporum f. sp. asparagi pathogenicity or response to climatic variables" submitted to Journal of Fungi. 

Comment 1: The authors describe the phylogenetic relationships between Foa isolates collected in northern and southern Europe, they test if (selected) isolates respond differently to experimental water and temperature manipulations in bioassays and experimentally show the effects of the on two variaties of asparagus plants . As an ecologist I especially found the combinations of field observations and manipulations in the lab interesting and meaningful, while I expected other methodologies based upon the title and I have some open questions concerning the analysis of the data. Please find some comments to further improve the manuscript below.

Reply: Thank you very much for your time and effort to review our document, we believe that you have improved the quality of our work. Here's a point-by-point response to your comments with some explanations:

Comment 2: It would be very helpful to the reader to formulate your three research questions at the end of the introduction. What are your expectations and do the results answer these questions? For example, is the genetic diversity of the fields assessed, or do you mean phylogenetic relatedness? What is pathogenetic diversity? shortly describe the experimental approach you use in the introduction and explain why you choose to test the pathogenicity of each of the isolates on 2 cultivars (explain the difference between the cultivars)

Reply: We have modified the end of the introduction to help the reader as suggested by the reviewer. On the other hand, the selection of the varieties has been made on the basis that they are two very widespread varieties in the main growing areas and with different susceptibility, it has been indicated in the text. The phylogenetic relationships for housekeeping gene fragments were used as a (conservative) proxy for intraspecies variability and we have clarified this in the text.

Comment 3: I also struggle a bit with the description of the methods. There seems to be no table 1 and table 2 comes after table 3 which is quite confusing for the reader. Please justify why you used partial sequences to construct the phylogenetic tree, are these primers not chosen for their conservativeness? Is it possible that the isolates differ more strongly in more functional parts of the genomes? How do the six selected isolates for the climate responses relate to the two clusters found in the phylogenetic tree?

Reply: We have included table 1, renumbered the tables 2 and 3 and placed them to facilitate the reading and understanding of the article. The primers for amplification of partial sequences of housekeeping genes (RPB1, RPB2, TEF1A) were chosen on basis of their consistent reliability on multiple fusarial isolates in our respective institutional collections. In fact, they were successfully used in previous works on strains from these collections (Galvez et al. 2017, Stępień et al. 2011). It is, of course, likely that the genomes differ more strongly in parts under weaker selective constraints (functional or non-functional) and with future next-generation sequencing efforts we hope to investigate this component of intraspecies diversity as a whole. The Discussion section was revised to take this consideration into account.

Comment 4: I’m not familiar with the software used in this manuscript but from the description it remains unclear which predictors were included in the models and how the model was structured. It would be good to have a table with a statistical summary of the model -including origin (north south), temperature and water availability as fixed predictors and isolate as random factor on mycelium growth (response). I cannot understand what Figure 2 is showing and where these extremely high R2 values come from. Fig 4 is already clearer, showing increased growth rates with increasing water activity. To show interactions between the isolate origin and temperatures one could colour the isolates from the north blue and from the south orange. It would be good to also have a summary table for the tests of pathogenicity on asparagus plants. These probably should include fixed effects and isolate random and the responses could be DSI and dry weight.

Reply:  We have improved the materials and methods section for a better understanding, figure 2 shows the average mycelial growth of the 6 isolates, the curves are trend lines adjusted to a second degree polynomial equation, but we understand that it can generate confusion for the reader and it does not provide any relevant information, so we have decided to remove them from the revised version. In this first figure it is intended to show the growth of all the isolates as a whole to be able to see the effect of the different treatments at a single glance, in figure 3 it is specified for each condition of temperature and water activity and difference between the origins of the isolates where the analysis of variance is performed for each of the cases and the significance is indicated (p-statistic). We understand that the information in figure 3 is actually more complete but we believe that figure 2 gives an initial global vision that helps to better understand the response to the different conditions and allows us to particularize in the following table.

We have colored the northern isolates blue and southern green as requested by the reviewer for a better understanding of Figure 4. The final part of Table 3 specifies the analysis of variance performed for the study of DSI and dry weight in function of the origin of the isolates and of the inoculated variety (Grande and Vegalin), where statistical significance is also indicated. 

Comment 5: In the discussion you mention that the genetic diversity of the tested pathogen is low resulting in similar responses along climatic gradients and similar disease effects of the isolates on asparagus plants. I tend to disagree with this, as you show the two clusters in the analysis as well as different responses for the isolates. I would strongly encourage the authors to redo the analysis and include origin into the models and resulting graphs. All in all I believe the study is topical and interesting but the manuscript needs some improvements before it can be plublished.   

Reply: As can be seen in figure 3, the dependence of mycelial growth rate on geographic origin of the isolates was analyzed and it has not yielded statistically significant differences (P> 0.05). So no direct correlation was detected. The same occurs with the experimental results of the pathogenicity tests, where no significant differences have been identified between the isolates from northern and southern Europe in any of the parameters evaluated (DSI of the aerial part and roots and dry weight of the part aerial and roots).

Reviewer 2 Report

General comments for the authors:

Article ID jof-1443396 entitled "Geographical origin does not modulate Fusarium oxysporum f. sp. asparagi pathogenicity or response to climatic variables" submitted to Journal of Fungi. 

Thanks for the opportunity to review this manuscript.  I think that this work makes an important contribution to understanding the response of Fusarium oxysporum populations to climatic variables.  I think several issues need to be tackle before its publication in the Journal of Fungi but it is just a matter to work a little bit on it.

Comment 1: The manuscript entitled " Geographical origin does not modulate Fusarium oxysporum f. sp. asparagi pathogenicity or response to climatic variables " has merit and can be published soon but there are some issues to tackle first.

Comment 2 (title): First of all, my biggest concerned is about the asseveration that Fusarium oxysporum f. sp. asparagi was the main fungi isolated from all the collected samples. To affirm that issue the authors must had done some host pathology tests with different plant families or at least showed a specific phylogenetic analyze that can be used as diagnosis. I am worry about it because according to some studies we have done, we have found cross pathogenicity of different formae specialis of Fusarium in different hosts, and also in the literature this phenomenon is well documented.

Comment 3 (abstract): I know there are different writing styles, but I would suggest to the authors to rewrite the abstract avoiding so long introduction and taking advantage of their interesting results including part of the discussion or conclusions in this section. This would bring them a plus for them and for the journal regarding the reading of this paper.

Comment 4 (abstract line 22 and 25): Replace the word “diversity” by “variability” or other one that fit better to what you analyze. The word diversity has a broad and complex meaning that authors did not tackle in this study. Author’s just analyze the pathogenicity and genetic variation of different Fusarium population from three countries but not the diversity of this fungi.

Comment 5 (introduction section ): I think this section can be reduced but again it depends of the writing and journal style.

Comment 6 (materials and methods section): The only missing issue here Is the one mentioned before regarding to explain more details regarding the diagnostic of F. o. f. sp. asparagi.

Comment 7 (materials and methods section): On the other hand, I would like that the authors justify why they use only six isolates from the 46 collected in the pathogenicity tests and why they do not use anyone from the Netherlands in the experiments.

Comment 8 (results section, lines 258-271): I think the authors get confused in these lines of this section because they are mixing issues of the discussion section in these lines. They need to separate these Ideas and placed in the properly section.

Comment 9 (results section, figure 2): Please add what does “aw” mean in this figure, and check whether the use of the symbol “,” in the numbers is okay according with this journal.

Comment 10 (results section, figure 2): Please add what does “aw” mean in this figure.

Comment 11 (discussion section, lines 331-332): Rewrite this sentence because this is not according with the title and the main objective of this study. It seems like this manuscript was done only to analyze genetic diversity.

Comment 12 (discussion section): Again, replace the word diversity for other one that fit better to each particular issue to develop or tackle.

Comment 13 (discussion section): Standardize the way you will call to Fusarium along the manuscript. In some parts you call it in one way and in other parts in different way.

Comment 14 (discussion section): The relative low genetic variation found it among Fusarium populations and the high adaptation of these populations to different environments can be explained by the genetic plasticity of this fungi. So you can have a look in the literature about this issue and include something of this genetic phenomenon to enrich the discussion.

Comment 12 (conclusion): Again, take care regarding the way you start the paragraphs because it seems like you are talking about other study because you begin with ideas not related to the main objectives of this study. At least this is what I think. So please rewrite this section.

Author Response

Rewiewer 2

Article ID jof-1443396 entitled "Geographical origin does not modulate Fusarium oxysporum f. sp. asparagi pathogenicity or response to climatic variables" submitted to Journal of Fungi. 

Thank you very much for your time and effort to review our document, we believe that you have improved the quality of our work.

Here's a point-by-point response to your comments with some explanations:

Comment 1: The manuscript entitled " Geographical origin does not modulate Fusarium oxysporum f. sp. asparagi pathogenicity or response to climatic variables " has merit and can be published soon but there are some issues to tackle first.

Comment 2 (title): First of all, my biggest concerned is about the asseveration that Fusarium oxysporum f. sp. asparagi was the main fungi isolated from all the collected samples. To affirm that issue the authors must had done some host pathology tests with different plant families or at least showed a specific phylogenetic analyze that can be used as diagnosis. I am worry about it because according to some studies we have done, we have found cross pathogenicity of different formae specialis of Fusarium in different hosts, and also in the literature this phenomenon is well documented.

Reply: We fully understand the reviewer's concern, this work is part of a doctoral thesis that will also try to clarify, among others, the parasitic specificity of isolates in different hosts. Among these tests, although we have not inoculated on other species yet, we have inoculated on asparagus all the isolates used in this study together with isolates of other special forms (Fusarium oxysporum f. sp. lycopersici, melonis and lactucae) only the asparagus isolates were pathogenic on asparagus. Although, as the reviewer indicates, inoculation in other species would allow us to identify this specificity, the fact that the isolates come from plants grown in fields with more than 7 years of monoculture and they have been isolated from the vascular system of the plant, which allows us to be reasonably sure of that they are the vascular form of the pathogen. The special form is collected in the scientific literature, in fact the previous work of our group already uses that denomination, although we fully understand the comment of the reviewer and we have modified the title to avoid identifying isolates as special forms. We have also modified the text where necessary throughout the manuscript.

Comment 3 (abstract): I know there are different writing styles, but I would suggest to the authors to rewrite the abstract avoiding so long introduction and taking advantage of their interesting results including part of the discussion or conclusions in this section. This would bring them a plus for them and for the journal regarding the reading of this paper.

Reply: We have modified the text according to the reviewer's suggestion.

Comment 4 (abstract line 22 and 25): Replace the word “diversity” by “variability” or other one that fit better to what you analyze. The word diversity has a broad and complex meaning that authors did not tackle in this study. Author’s just analyze the pathogenicity and genetic variation of different Fusarium population from three countries but not the diversity of this fungi.

Reply: We have modified the text according to the reviewer's suggestion.

Comment 5 (introduction section ): I think this section can be reduced but again it depends of the writing and journal style.

Reply: Thank you for the comment, we have shortened the introduction, however we believe that most of issues commented in the introduction is pertinent to understand the real problem that this work addresses and the consequences for the asparagus producing sector. Of course we follow the journal's instructions, but if the editor also finds it too long, we may shorten its content.

Comment 6 (materials and methods section): The only missing issue here Is the one mentioned before regarding to explain more details regarding the diagnostic of F. o. f. sp. asparagi.

Reply: Addressed in answer to comment 1.

Comment 7 (materials and methods section): On the other hand, I would like that the authors justify why they use only six isolates from the 46 collected in the pathogenicity tests and why they do not use anyone from the Netherlands in the experiments.

Reply: We have selected six isolates because we believe that it is an acceptable number when we want to analyze 4 temperatures and 4 water activities, the number of plates analyzed is already very high. Regarding the isolates from the Netherlands, there is no scientific reason, we actually obtained them once the response tests for environmental factors were completed, but we decided to include them in the phylogenetic studies.

Comment 8 (results section, lines 258-271): I think the authors get confused in these lines of this section because they are mixing issues of the discussion section in these lines. They need to separate these Ideas and placed in the properly section.

Reply: Indeed we had mixed results with discussion as the reviewer points out, we have separated and repositioned it in its appropriate sections.

Comment 9 (results section, figure 2): Please add what does “aw” mean in this figure, and check whether the use of the symbol “,” in the numbers is okay according with this journal.

Reply: We have modified the text to include the meaning of aw according to the reviewer's suggestion.

 Comment 10 (results section, figure 2): Please add what does “aw” mean in this figure.

Reply: We have modified the figure according to the reviewer's suggestion.

Comment 11 (discussion section, lines 331-332): Rewrite this sentence because this is not according with the title and the main objective of this study. It seems like this manuscript was done only to analyze genetic diversity.

Reply: We have modified the text according to the reviewer's comment

Comment 12 (discussion section): Again, replace the word diversity for other one that fit better to each particular issue to develop or tackle.

Reply: We have replace the word as suggested.

Comment 13 (discussion section): Standardize the way you will call to Fusarium along the manuscript. In some parts you call it in one way and in other parts in different way.

Reply: We have tried to unify criteria throughout the entire manuscript, having changed the title we have also changed the reference to the special form when calling Fusarium.

Comment 14 (discussion section): The relative low genetic variation found it among Fusarium populations and the high adaptation of these populations to different environments can be explained by the genetic plasticity of this fungi. So you can have a look in the literature about this issue and include something of this genetic phenomenon to enrich the discussion.

Reply: We have also modified the text where necessary throughout the manuscript.

Comment 12 (conclusion): Again, take care regarding the way you start the paragraphs because it seems like you are talking about other study because you begin with ideas not related to the main objectives of this study. At least this is what I think. So please rewrite this section.

 Reply: We have included some revisions to the conclusion. We think that conclusions outline implications of this investigation, in particular for future efforts in developing modes of plant protection.

Round 2

Reviewer 1 Report

The authors have replied to the suggestions and improved parts of the manuscript; however, I feel there are some unanswered questions left. Maybe I was not clear enough and I haven’t convinced the authors that a little extra effort in an extended description and additional model could help to drastically improve this manuscript and our understanding of fusarium isolates. However, I have not found an statistical model to directly test if isolated from different origin react differently to water under certain temperatures this would require to test the three-way interaction between origin:water:temp . Similarly for the tests of pathogenicity on asparagus plants one should test the interaction between origin (north south) and commercial variety (Grande Vegalin) as proposed before. I thing if the correct statistical model will be applied one might be able to detect this difference statistically. Therefore, I would like to see a statistical summary table with the degrees of freedom, sums of squares F-values and P values for each of the predictors (main effects) and their interactions, or a very strong argument why not to do this.

Additionally i would strongly encourage the authors to extend the figure legends and table headers. Explain all colours, lines and abbriviations we see and add summary statistics here as well.

Thank you and good luck.

Author Response

Thank you for reviewing our article again, indeed we had not understood well the initial suggestion of the additional model to detect difference statistically. Actually, we had done the model, and no differences have come out, but we understand that including the results could help understanding of the manuscript and the absence of differences between the origins has to be indicated with the statistics. 

The result of the three-way ANOVA test interaction between origin-water-temperature is shown in the following table, as it can be seen, there are no statistically significant differences between the different origins, neither in second order or third order interactions.

Similar results are obtained when analyzing the tests of pathogenicity on asparagus plants. The results on the effect of origin and variety on the disease index in the root system (P=9353) and in the aerial part (P=0.7387) do not show significant differences either.  

There are already 8 tables and figures in the manuscript, so we decided not include another table, but he have included these data within the text (both for directly test if isolated from different origin react differently to water activity under certain temperatures and for patogenicity tests with regard interaction between varieties and origins). We have also improve the figure legends to explain colors and abbreviations.
